# Dupilumab Leads to Clinical Improvements including the Acquisition of Tolerance to Causative Foods in Non-Eosinophilic Esophagitis Eosinophilic Gastrointestinal Disorders

**DOI:** 10.3390/biom13010112

**Published:** 2023-01-05

**Authors:** Naoya Arakawa, Hisako Yagi, Mariko Shimizu, Daisuke Shigeta, Akihiko Shimizu, Shigeru Nomura, Takumi Takizawa, Yoshiyuki Yamada

**Affiliations:** 1Department of Pediatrics, Gunma University Graduate School of Medicine, 3-39-22 Showa-machi, Maebash 371-8511, Gunma, Japan; 2Division of Allergy and Immunology, Gunma Children’s Medical Center, 779 Shimohakoda, Hokkitsu, Shibukawa 377-8577, Gunma, Japan; 3Department of Pediatrics, Saku Central Hospital Advanced Care Center, 3400-28 Nakagomi, Saku 385-0051, Nagano, Japan; 4Department of Pediatrics, Tokai University School of Medicine, 143 Shimokasuya, Isehara 259-1193, Kanagawa, Japan

**Keywords:** dupilumab, biologic, eosinophil, eosinophilic gastrointestinal disorder (EGID), non-eosinophilic esophagitis (EoE) eosinophilic gastrointestinal disorder (EGID) (non-EoE EGID), elimination diet

## Abstract

A recent report showed that most pediatric cases of non-eosinophilic esophagitis (EoE) eosinophilic gastrointestinal disorders (EGIDs) (non-EoE EGIDs) are persistent and severe compared with those of EoE, thus requiring further effective therapeutic approaches. In this study, we present the first case based on a systematic search of non-EoE EGID for which tolerance to causative foods and histological and symptomatic improvements were achieved following dupilumab administration, after elimination diets and omalizumab and mepolizumab treatments. Driven by this case, we investigated the efficacies of biological treatments in non-EoE EGID cases based on the patient studied herein, and other patients identified in the conducted systematic review. Seven articles, including five different biologics, were reviewed. Both clinical efficacies and impact differences among the targeted molecules are demonstrated in this study. Our findings show that dupilumab may affect mechanisms that can suppress symptoms induced by offending foods that are different from those induced by other biologics as identified in the conducted systematic review. Additional studies are required to address the unmet needs of non-EoE EGID treatments.

## 1. Introduction

Eosinophilic gastrointestinal disorders (EGIDs) are diseases characterized by allergic and eosinophilic inflammation of the gastrointestinal (GI) tract and are classified into two categories: eosinophilic esophagitis (EoE), and EGID other than EoE, recently defined as non-EoE EGID [1,2]. Non-EoE EGIDs are rare in Western countries, unlike EoE [3]; however, they are more common in Japan [4], especially in children [5]. In 2021, a large-scale, national Japanese survey revealed that most pediatric cases of non-EoE EGIDs were persistent and severe compared with those of EoE and require additional effective therapeutic options [5]. Systemic glucocorticoids are the most widely used treatment for non-EoE EGIDs. However, two-thirds of treated patients become glucocorticoid-dependent or develop resistance [1,6]. In addition, elimination diets, including empirical elimination [7,8] or elemental diets [9], can often be promising therapeutic approaches. However, the efficacy of these dietary modifications remains undetermined, as cases with symptomatic improvements have been primarily described in case reports thus far [10]. Moreover, although a randomized clinical trial revealed that montelukast was effective for pediatric patients with dyspepsia and duodenal eosinophilia [11], this may be only useful for patients with mild forms of these conditions. The other promising treatment is biological therapy for molecules associated with the Type 2 pathway. Among the biologics, mepolizumab [12,13], benralizumab [14], omalizumab [13,15], reslizumab [14], dupilumab [16], lirentelimab [17], and cendakimab [18] are associated with Type 2 inflammation. In addition, vedolizumab is an antibody against α_4_β_7_ integrin that is exclusively expressed on leukocytes in the GI system, such as eosinophils and T lymphocytes [19,20,21]. Thus, we hypothesized that these biologics may be effective against non-EoE EGIDs.

To our knowledge, we report herein the first case of non-EoE EGID that achieved not only symptomatic and histological improvements but also tolerance to the foods responsible for non-EoE EGID following the administration of dupilumab. This patient was treated initially with elimination diets, and subsequently with three biologics (available for asthma in Japanese children) [22]. An initial search found only a case report of non-EoE EGIDs associated with the efficacy of dupilumab. Therefore, we expanded the investigation to several biologics (as mentioned above) and considered a comparison of the effects of each therapy based on a discussion supported by a systematic review.

## 2. Materials and Methods

### 2.1. Patients

Patients with non-EoE EGIDs treated with the biologics described above were included; these were identified from a systematic literature review using the undermentioned strategy. In addition, our case was also included for analysis purposes: a 15 year-old girl presented with frequent diarrhea and vomiting at the age of 7 years, and was diagnosed with non-EoE EGID. The clinical diagnosis of this case was confirmed based on the criteria used in a Japanese national survey [5]. In the Gunma prefecture, which has two million inhabitants, patients with pediatric EGIDs who required advanced therapies, such as elimination diets and/or biologics, were referred to either the Gunma University or Gunma Children’s Medical Center, two tertiary pediatric units. To our knowledge, this case presents the only patient with EGID who was treated with all three biologics available for Japanese children in our area. This study was approved by the Ethics Committee of Gunma Children’s Medical Center (GCMC2018-34). Written informed consent was obtained from the patient for publication of the case study.

### 2.2. Literature Review of Biologics Use for Non-EoE EGIDs

A systematic literature search of PubMed was performed by N.A. and Y.Y. using updated search criteria based on the published Japanese guidelines for EGIDs. The guidelines were written in Japanese, but search queries were available in English [23]. Studies associated with primary non-EoE EGIDs were identified based on a comprehensive search of the Japanese guidelines for EGIDs conducted previously by a literature search expert at the medical library. In this study, the search period was extended to 22 August 2022, and Japanese articles, including those with English abstracts, were removed. In addition, the selected studies were only associated with biologics, such as mepolizumab, benralizumab, omalizumab, reslizumab, dupilumab, cendakimab, lirentelimab, and vedolizumab. Additionally, studies related only to eosinophilic esophagitis or EGID—secondary to other eosinophilic diseases—were excluded from the search. The search strategy and flowchart of the literature search are shown in Table 1 (search results were downloaded from PubMed Advanced Search Builder) and Figure 1, respectively.

## 3. Results

### 3.1. Literature Review

The search strategy yielded 44 references. Additionally, 28, 3, 1, 1, and 1 references were only associated with EoE, inflammatory bowel diseases, other eosinophilic diseases, non-eosinophilic diseases, and a biologic, respectively, and were thus excluded. Among the remaining 10 references related to non-EoE EGIDs, three were excluded because they were not related to biologic therapy. In total, seven articles were included in this review.

Omalizumab, mepolizumab, vedolizumab, dupilumab, and lirentelimab were respectively used in two, two, two, one, and one studies, which consisted of 10, 3, 8, 3, and 65 patients, respectively (both omalizumab and mepolizumab were used in one patient) (Table 2). In addition to these patients, our case, which was treated with omalizumab, mepolizumab, and dupilumab, is included in Table 2.

Two prospective studies, a randomized double-blind placebo-controlled and a single-center open-label, were performed using lirentelimab [17] and omalizumab [15], respectively. In patients with eosinophilic gastritis (EoG) or eosinophilic duodenitis (EoD), lirentelimab significantly reduced GI eosinophils and symptoms compared with a placebo [17]. In contrast, omalizumab reduced tissue eosinophils, but the reduction was not statistically significant for EoG and EoD. However, the symptom scores decreased significantly [15].

In terms of vedolizumab, when combined with two retrospective studies [20,21], changes in GI eosinophils from baseline were evaluated in eight patients with non-EoE EGIDs. No or very few tissue eosinophils were observed in one patient at baseline and after treatment based on the graph in the respective article [20]. The remaining seven patients demonstrated histological improvements. Nevertheless, among the nine patients evaluated herein, three did not exhibit symptomatic improvements, and vedolizumab was partially effective in two, thus indicating that vedolizumab may have a limited effect against non-EoE EGIDs.

Mepolizumab was used in two studies [12,13]; three patients were treated. Only one achieved complete histological improvement, while another patient in the same study showed increased tissue eosinophils following mepolizumab treatment. However, overall clinical improvements were observed in all three patients.

More recently, three patients with non-EoE EGID who had been refractory to standard therapies or exhibited significant side effects due to the swallowing of steroids were treated with dupilumab [16]. Dupilumab administration was associated with histological and symptomatic improvements in all three patients.

One patient presented in a case report had been treated with both mepolizumab and omalizumab [13]. However, the patient’s symptoms improved immediately after mepolizumab treatment, and omalizumab was administered for mepolizumab-refractory asthma. Therefore, it could not be determined as to whether omalizumab was effective for non-EoE EGID in this patient.

All selected studies and our case were associated with EoG. However, the only cases with eosinophilic colitis (EoC) requiring colonoscopy for diagnosis were two patients in one study and our case. This may be because upper GI endoscopy is often performed when EoE, which is more common in Western countries, is suspected, and the stomach and duodenum are immediately examined.

No reports were identified regarding the use of reslizumab, benralizumab, or cendakimab related to primary non-EoE EGIDs in this search strategy. Therefore, additional searches for these three biologics were performed using PubMed and clinicaltrials.gov (https://clinicaltrials.gov, accessed on 29 August 2022) to determine if any published studies could be extracted. The results are outlined below. Reslizumab as SCH55700 has been used for patients with EoG and led to clinical improvements. However, this study was performed to identify the mechanism underlying rebound eosinophilia after treatment (just a single dose of SCH55700) [24]. The efficacy of benralizumab associated with EGIDs secondary to hypereosinophilic syndromes has been reported, although some patients had recurrent symptomatic flares [25]. In addition, there are ongoing clinical trials that use benralizumab (NCT03473977) or cendakimab (NCT05214768) for non-EoE EGIDs.

### 3.2. Dupilumab Led to the Acquisition of Tolerance to Causative Foods in Non-EoE EGID

In our case, dupilumab administration led to the acquisition of tolerance to foods responsible for non-EoE EGID. A 15 year-old girl presented with frequent diarrhea and vomiting at the age of 7 years and was diagnosed with non-EoE EGID, including EoG, EoD, and EoC based on GI endoscopy (Figure 2A,B). Non-EoE EGID was associated with protein-losing enteropathy. Hen’s eggs and cow’s milk consumed daily in school lunches were determined as causative foods and were subsequently removed; this elimination improved the symptoms and intense peripheral blood eosinophilia, but severe GI eosinophilia persisted. Of note, GI eosinophil counts were high, even when we assessed the tissue eosinophil count using recent criteria for non-EoE EGID in consideration of eosinophils physiologically present in the GI tract [5,26]; additional foods were eliminated empirically, and resulted in additional improvements. In the food reintroduction process, cow’s milk was found to be the cause. The patient maintained symptomatic remission of non-EoE EGID by avoiding possible offending foods. At the age of 11 years, she began taking omalizumab for poorly controlled asthma; however, owing to the inadequate response, her medication was switched to mepolizumab at the age of 12 years. Two years later, her asthma symptoms worsened, and she was started on dupilumab when she was 15 years old. Both the asthma and its symptoms improved, and an oral reintroduction confirmed that the patient had developed tolerance to hen’s eggs and cow’s milk (Figure 2A). The successful reintroduction seemed to be due to the effect of dupilumab, since she had occasionally presented symptoms after accidental consumption of cow’s milk or hen’s eggs during omalizumab and mepolizumab treatments. Anti-immunoglobulin E (IgE) antibodies and the suppression of the interleukin (IL)-5 signaling pathway improved the endoscopic findings and chronic digestive symptoms of non-EoE EGID; nevertheless, acute and subacute symptoms had occurred due to ingestion of the causative foods, meaning that the improvements were not clinically sufficient, and suppression of the IL-4/IL-13 signaling pathways led to the acquisition of tolerance to the causative foods.

## 4. Discussion

Biologics available for Type 2 inflammation include mepolizumab and reslizumab (anti-IL-5 antibodies) [14], benralizumab (anti-IL-5 receptor alpha antibody) [14], omalizumab (anti-IgE antibody) [13,15], dupilumab (anti-IL-4 receptor alpha antibody) [14,16], cendakimab (anti-IL-13 antibody) [14,18], and lirentelimab (anti-siglec-8 antibody) [17]. Among these seven biologics, five were used in patients with primary non-EoE EGIDs in the seven studies included in this review. In contrast to Type 2 inflammation, vedolizumab is a humanized antibody that binds to a conformational epitope of α_4_β_7_ integrin expressed on gut-tropic eosinophils as well as on T lymphocytes. The clinical efficacy of vedolizumab in patients with inflammatory bowel diseases was demonstrated previously [19]; two studies reported the clinical efficacy of vedolizumab in non-EoE EGIDs in this review [20,21].

Among all the biologics reviewed herein, lirentelimab was the only one that exhibited significant histological and symptomatic improvements in patients with non-EoE EGIDs, as proven by a well-designed, high-quality study [17]. However, a press release form of a Phase III study (NCT04322604) showed a significant histologic response with no symptomatic improvements.

In 2007, an open-label study was performed for non-EoE EGIDs using omalizumab [15]. Although it was not significant, a downward trend in gastric and duodenal eosinophilia was observed, and different disease indices from the current ones had been used; symptoms were statistically improved following omalizumab treatment.

Two previous case reports on mepolizumab exist, including three patients with non-EoE EGIDs [12,13]. Although clinical improvements were observed in all three patients, one patient did not undergo histological evaluation after treatment, and histological improvements were achieved in one case; surprisingly, the other patient showed exacerbation of tissue eosinophilia [12]. Although well-designed studies have been performed using mepolizumab in EoE, unlike non-EoE EGIDs, significant histological but not symptomatic improvements have been demonstrated [27,28]. The efficacy of mepolizumab in EGIDs may be equivocal.

Recently, dupilumab was approved by the United States Food and Drug Administration for the treatment of EoE that resulted in the first FDA approval of a treatment for EoE [29]. Therefore, dupilumab may offer promise as an approach, even for non-EoE EGIDs. However, except for our case, only one report showed histological and symptomatic improvements in three patients with intractable non-EoE EGIDs [16], but the relevant clinical trial is ongoing (NCT03678545).

Apart from Type 2 inflammation, vedolizumab selectively affects GI inflammation and has a favorable safety profile. Unfortunately, only four out of nine patients in two studies included in this review exhibited apparent clinical improvements [20,21]. Thus, additional research is required to determine the indications for vedolizumab treatment.

Interestingly, the efficacy of elimination diets and the three different biologics reviewed above were observed in the patient case studied herein. Although the patient has also been treated with systemic steroids during asthma exacerbation, the therapy for non-EoE EGID seems to have produced minimal effects because it was temporary. No therapies specific to EGIDs have yet been covered by health insurance in Japan. Therefore, biologics may only be used to treat health insurance-covered diseases, such as asthma and atopic dermatitis [22]. Therefore, the use of these three biologics in EGIDs is rare. First, the elimination of possible causative foods improved the symptoms and severe peripheral blood eosinophilia suggestive of hypereosinophilic syndromes, but severe GI eosinophilia remained. Additional empirical, multiple food-elimination diets induced minor additional improvements, and a subsequent reintroduction phase (oral challenge tests on consecutive days) revealed cow’s milk as a causative food. Furthermore, omalizumab was introduced and switched to mepolizumab for uncontrolled severe asthma. Both led to additional improvements in non-EoE EGIDs, but these improvements were not observed in the case of asthma (Appendix A). Eventually, switching to dupilumab improved asthma symptoms. Surprisingly, the eliminated foods (e.g., hen’s eggs and cow’s milk) were safely reintroduced, which resulted in an asymptomatic state for the patient. This may be explained by the fact that EGID is regarded as a mixed type of allergy, with a mechanism exhibiting combined features based on IgE- and non-IgE-mediated mechanisms [30]. Specifically, dupilumab may suppress not only eosinophilic inflammation but also an IgE-mediated mechanism. Genome-wide transcript profiles of EoG showed that *IL4* and *IL13* were highly upregulated in EoG compared with control gastric tissue, and *IL13* expression correlated significantly with *CCL26* (*eotaxin-*3) expression [31]. Spekhorst et al. reported a profound IgE-mediated decrease in specific IgE levels of food allergens during dupilumab treatment in patients with food-allergic atopic dermatitis [32]. Taken together, dupilumab may be a more definitive therapy.

This study is associated with some limitations. First, we reported only one case in conjunction with a systematic review derived from only seven publications. Second, because biologics are expensive drugs, they may only be used when covered by health insurance; therefore, it is difficult for physicians to experience and report more cases. Comparing the effects of the three biologics used in this non-EoE EGID patient was difficult because all three drugs were effective. Therefore, the study focused on the acquisition of tolerance to offending foods responsible for non-EoE EGID. This may be an interesting hypothesis to evaluate these biologics, although a head-to-head comparative trial among them may be proven difficult. When viewed in a different light, evaluation using GI eosinophil counts may be inconclusive because of the presence of physiological eosinophils. Referring to the studies on EoE, assessment using immunohistochemical markers, such as IgG4, eotaxin-3, cysteine-rich secretory protein 3, GATA binding protein 3, and periostin matrix metalloproteinases, can be useful in disease diagnosis and determination of appropriate treatments [33].

In conclusion, all five biologics used for non-EoE EGIDs in the reviewed literature were effective at varying degrees. Although it appears difficult to determine which of these biologics was most effective, our case may imply that dupilumab is affected by mechanisms that can suppress symptoms induced by offending foods that are different from those induced by omalizumab and mepolizumab. In terms of the molecular mechanisms of non-EoE EGIDs, the unique effect of dupilumab in the acquisition of tolerance to causative foods in this case indicates that non-EoE EGIDs may be affected by both IgE- and non-IgE-mediated mechanisms, suggesting a mixed type of allergy. Since this study lacks sufficient information to make relevant conclusions, future, well-designed studies are required to address the unmet treatment needs for persistent and severe non-EoE EGIDs.

## Figures and Tables

**Figure 1 biomolecules-13-00112-f001:**
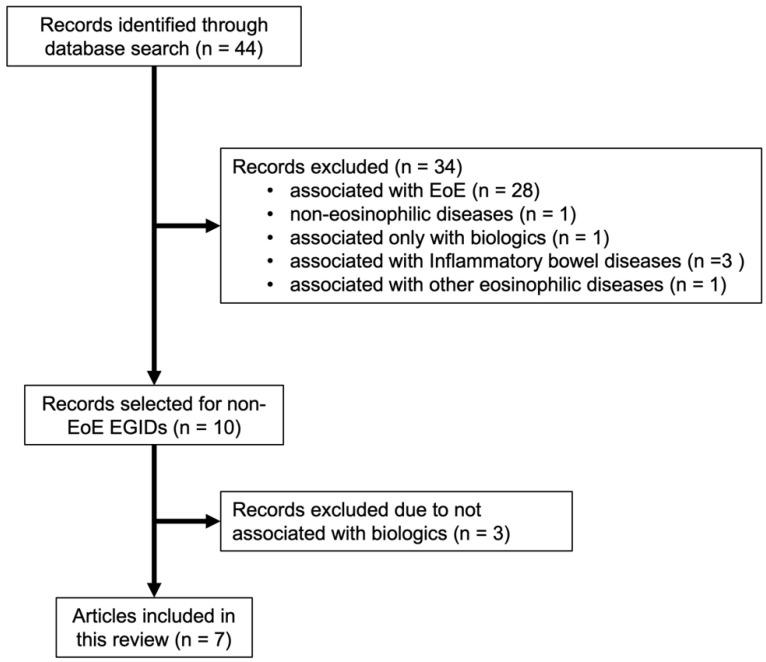
Flowchart of the literature search (EoE, eosinophilic esophagitis; EGID, eosinophilic gastrointestinal disorder).

**Figure 2 biomolecules-13-00112-f002:**
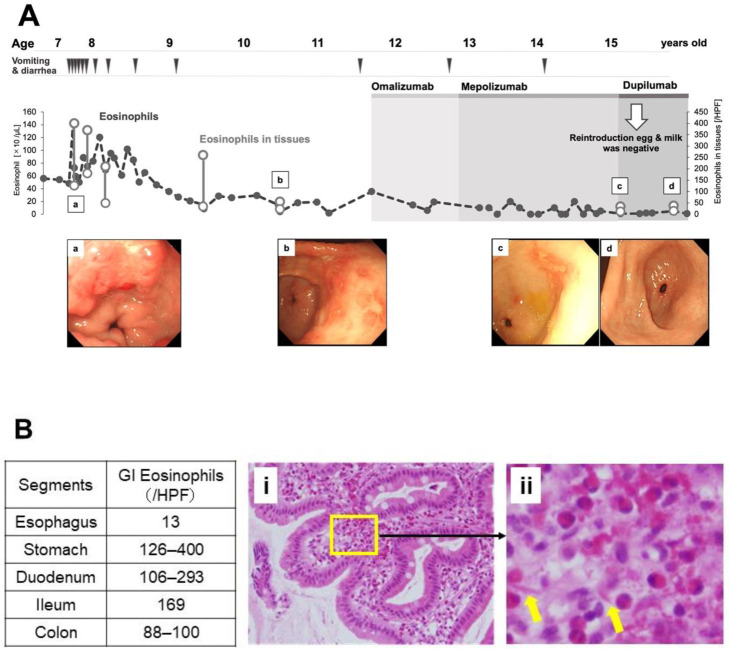
Clinical course of the patient with non-EoE EGID. (**A**) Peripheral blood (line chart) and gastric (vertical line) eosinophil counts and endoscopic findings of a patient’s stomach (a–d) are shown in chronological order. (**B**) Histopathologic analysis of the patients. Gastrointestinal eosinophil counts in each segment are tabulated. (**i**) Histological findings of the duodenum using hematoxylin and eosin staining (optical magnification 200×). (**ii**) The inset shows a magnified view of the original image. The arrows point to Charcot–Leyden crystals. HPF, high-power field; GI, gastrointestinal.

**Table 1 biomolecules-13-00112-t001:** Search strategy used for the systematic reviews conducted in this study (PubMed).

Search Number	Query	Results
#1	((“Eosinophilic enteropathy”[TW] OR “Eosinophilic gastrointestinal disorders”[TW] OR “Eosinophilic gastroenteritis”[TW] OR “Eosinophilic enteritis”[TW] OR “Eosinophilic duodenitis”[TW] OR “Eosinophilic colitis”[TW] OR “Gastrointestinal eosinophilia”[TW] OR “Intestinal eosinophilia”[TW] OR “Gastric eosinophilia”[TW] OR “Colonic eosinophilia”[TW] OR “Intestinal eosinophil infiltration”[TW] OR “Gastric eosinophil infiltration”[TW]) OR (“Eosinophilic enteropathy”[Supplementary Concept]) OR ((eosinophilia[Mesh]) AND (Gastroenteritis[MH] OR “Intestinal Diseases”[MH] OR “Stomach Diseases”[MH]))) AND “humans”[MeSH Terms] AND (“child”[MeSH Terms] OR “adolescent”[MeSH Terms] OR “adult”[MeSH Terms]) AND 1970[PDAT]: 2022[PDAT] AND (English[LA]) AND (“Treatment Outcome”[MH] OR “therapeutics”[MeSH Terms] OR “therapy”[Subheading])	1337
#2	mepolizumab OR benralizumab OR omalizumab OR reslizumab OR dupilumab OR lirentelimab OR vedolizumab OR cendakimab	7604
#3	#1 AND #2	44

**Table 2 biomolecules-13-00112-t002:** Studies using biologics in non-EoE EGIDs.

Articles	Biologics	Study Population	Age Group	Histological Change	Symptomatic Improvement	Segments	Study Design	Year
Dellon et al. [17]	Lirentelimab	n = 65	18–80	change in GI Eos ^†^ (mean): placebo (−9%), active drug (−86%) ^1^	change in total symptom score ^†^: placebo (−22%), active drug (−48%) ^1^	EoG	randomized, double-blind, placebo-control study	2020
Foroughi et al. [15]	Omalizumab	n = 9	12–76	change in GI Eos ^†^ (mean): gastric antrum (−69%) and duodenum (−59%)	change in total symptom score ^†^ (−70%) ^1^	EoG, EoD	single-center open-label study	2007
Patel et al. [16]	Dupilumab	n = 3 ^2^	7, 14, and 9	change in GI Eos ^†^: stomach (−88%) (n = 2), duodenum (−81%), and jejunum (−15%) (n = 1)	symptomatic improvement (3/3 patients)	EoG, EoD, EoJ	case reports	2022
Grandinetti et al. [20]	Vedolizumab	n = 4 ^3^	13–78	change in GI Eos ^†^ (−66%) (n = 4) ^4^	symptomatic improvement 3/4 patients ^5^	EoD + EoEEGEEoCEoE + EoN + EoC	retrospective cohort analysis	2019
Kim et al. [21]	Vedolizumab	n = 4 ^3^	22.7–53.7	changes in GI Eos ^†^: stomach (n = 0), duodenum (−100%) (n = 2), and jejunum (−29%) (n = 1)	symptomatic improvement 3/5 patients	EoG + EGE EoE + EoG + EoN (n = 2) EoG + EoN	retrospective study	2018
Benjamin et al. [12]	Mepolizumab	n = 2	29.5 (mean)	change in GI Eos: stomach (−100% and 125%) (increased) (n = 2)	clinical improvement (overall) 2/2 patients	EoE + EGE (n = 2)	retrospective study	2018
Han and Lee [13]	Mepolizumab(+ omalizumab) ^6^	n = 1	67	no evaluation after treatment	clinical improvement: mepolizumab alone	EoE + EGE	case report	2018
Arakawa et al. (our case) ^7^	Omalizumab, mepolizumab and dupilumab	n = 1	15	maintenance of historical remission: all three drugs reduced GI Eos: omalizumab and mepolizumab endoscopic improvement: dupilumab	maintenance of clinical remission: all three drugs acquisition of tolerance to causative foods: dupilumab	EoG + EGE + EoC	this study	202

GI, gastrointestinal; Eos, eosinophils; EGID, eosinophilic gastrointestinal disorder; EoE, eosinophilic esophagitis; EoG, eosinophilic gastritis; EoD, eosinophilic duodenitis; EoN, eosinophilic enteritis; EoJ, eosinophilic jejunitis; EGE, eosinophilic gastroenteritis; EoC, eosinophilic colitis. ^†^ percentage change from baseline. ^1^ Statistically significant. ^2^ Two patients were refractory to standard therapies, and one had significant side effects upon swallowing steroids. ^3^ Steroid-refractory or -dependent. ^4^ Peak eosinophil counts (high-power field) were estimated based on the figure in the article. ^5^ Two patients showed partial improvements. ^6^ EGE was effectively controlled by mepolizumab alone, and omalizumab was added due to mepolizumab-refractory asthma. ^7^ Details are included in the main text.

## Data Availability

Not applicable.

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
