# Peer review of "Dupilumab Leads to Clinical Improvements including the Acquisition of Tolerance to Causative Foods in Non-Eosinophilic Esophagitis Eosinophilic Gastrointestinal Disorders"

_biomolecules, 2023, doi:10.3390/biom13010112_

Round 1

Reviewer 1 Report

Arakawa et al. investigated the efficacies of biological treatments in non-EoE EGID. Despite some issues, this study is interesting and offers new information on non-EoE EGID treatment.

1.        Introduction

Please indicate the non-EoE EGID diagnostic criteria.

Please describe in more detail the treatment strategies for non-EoE EGID.

The authors already in the title suggest the benefits of using dupilumab in the treatment of EGID non-EoE. There is no information about the mechanism of its action.

2.       Results.

The analysis includes a small number of articles, so  is it impossible to draw any conclusions on this basis?

Based on the introduction, it can be assumed that the article will be about the pediatric group, but the articles in the review also apply to adults. Please add information in Table 1 about the age of the patients included in the study.

Line 146: Omalizumab is capitalized. Why?

Was this patient from case report in need of systemic steroids (due to lack of asthma treatment effect)? Provide this information. If so, can steroid use affect the effect of  non-EoE EGID therapy?

3.       Discussion

The authors rightly pointed out that it is difficult to generalize the obtained results on the basis of one patient.

Based on the analysis of the literature, do the authors see a relationship with the affected gastrointestinal tract (gastritis, duodenitis, etc.) and the treatment effect?

Could other diagnostic methods be useful in the treatment of non-EoE EGID? For example, immunohistochemistry? Perhaps the observations obtained in patients with EoE could be used for further research in non-EoE patients. Suggested reference: doi: 10.1080/00365521.2020.1831053

Author Response

Dear Reviewer

We thank reviewers for careful reading our manuscript and for giving us fruitful suggestions. Based on the reviewers’ comments, we have revised our manuscript entitled “Dupilumab Leads to Clinical Improvements Including the Acquisition of Tolerance to Causative Foods in Non- eosinophilic Esophagitis Eosinophilic Gastrointestinal Disorders” by Naoya Arakawa et al. I feel that the revised manuscript is significantly improved over the initial submission thanks to the reviewers’ suggestions. I would appreciate it if you would consider the manuscript for publication in Biomolecules.

Thank you in advance for your kind consideration of this paper.

Yoshiyuki Yamada

Point-by-point response for Reviewer 1 (R1)

Arakawa et al. investigated the efficacies of biological treatmentsin non-EoE EGID. Despite some issues, this study is interestingand offers new information on non-EoE EGID treatment.

We thank the reviewer for giving us important and pertinent comments.

Comment R1-1: Please indicate the non-EoE EGID diagnostic criteria.

Response R1-1: Since there is no established diagnostic criteria for non-EoE EGID, we used the criteria that was used in Japanese national survey (reference 5) in this time. We add sentence “The clinical diagnosis of case was confirmed based on the criteria used in Japanese national survey [5]” at line 75 in revised version (no tracking).

Comment R1-2: Please describe in more detail the treatment strategies for non-EoE EGID.

Response R1-2: Based on the reviewer’s suggestion, we added the sentences “Systemic glucocorticoids are the most widely used treatment for non-EoE EGIDs. However, two-thirds of treated patients become glucocorticoid dependent or develop resistance [6,7]. In addition,” at line 45 and “Moreover, although a randomized clinical trial revealed that montelukast was effective for pediatric patients with dyspepsia and duodenal eosinophilia [11], this may be only useful for patients with mild forms of these conditions.” at line 50 in revised version (no tracking).

Comment R1-3: The authors already in the title suggest the benefits of using dupilumab in the treatment of EGID non-EoE. There is no information about the mechanism of its action.

Response R1-3: Based on the reviewer’s suggestion, “Genome-wide transcript profiles of EoG showed that IL4 and IL13 were highly upregulated in EoG compared with control gastric tissue, and IL13 expression correlated significantly with CCL26 (eotaxin-3) expression [30]. Spekhorst et al. reported a profound IgE-mediated decrease in specific IgE levels of food allergens during dupilumab treatment in patients with food-allergic atopic dermatitis [31]. Taken together, dupilumab may be a more definitive therapy.” has been included at line 270 in revised version (no tracking).

Comment R1-4: The analysis includes a small number of articles, so is it impossible to draw any conclusions on this basis?

Response R1-4: I agree with the reviewer that it is difficult to conclude. As we discussed in limitations, because biologics are expensive drugs, they may only be used when covered by health insurance; therefore, it is difficult for physicians to experience and report more cases. Therefore, we expected a small number of articles about biologics. However, this literature review appeared to suggest that at least biologics are promising approach as a whole, since two higher evidenced studies has been included and some clinical trials are ongoing. We added a phrase “Since this study lacks sufficient information to make relevant conclusions,” in conclusion to emphasize it at line 297 in revised version.

Comment R1-5: Based on the introduction, it can be assumed that the article will be about the pediatric group, but the articles in the review also apply to adults. Please add information in Table 1 about the age of the patients included in the study.

Response R1-5: As we described, there are more unmet needs for pediatric non-EoE EGID. Therefore, we are more interested in pediatric group. However, there are small number of articles in the review and we included all ages. We added age groups in table 2.

Comment R1-6: Line 146: Omalizumab is capitalized. Why?

Response R1-6: We decapitalized it.

Comment R1-7: Was this patient from case report in need of systemic steroids (due to lack of asthma treatment effect)? Provide this information. If so, can steroid use affect the effect of  non-EoE EGID therapy?

Response R1-7: Since systemic steroids were used just for acute exacerbations, systemic steroids were administered within a week each acute exacerbation. Acute exacerbations were a few times per year even when uncontrolled as shown in Supplementary Figure 1. Therefore, systemic steroids use little affected as a therapy of non-EoE EGID. We added a sentence, “Although the patient has also been treated with systemic steroids during asthma exacerbation, the therapy for non-EoE EGID seems to have produced minimal effects because it was temporary.” At line 251 in revised version.

Comment R1-8: The authors rightly pointed out that it is difficult to generalize the obtained results on the basis of one patient.

Response R1-8: I totally agree with the reviewer. Therefore, we added as described in Response-R1-4 (line 297 in revised version).

Comment R1-9: Based on the analysis of the literature, do the authors see a relationship with the affected gastrointestinal tract (gastritis, duodenitis, etc.) and the treatment effect?

Response R1-9: Since our patient and only one study using vedolizumab was associated with EoC, it is difficult to compare the efficacies of biologics among the affected GI tract. We added a paragraph “All selected studies and our case were…are immediately examined.” (line 159-163 in revised version) to explain it.

Comment R1-10: Could other diagnostic methods be useful in the treatment of non-EoE EGID? For example, immunohistochemistry? Perhaps the observations obtained in patients with EoE could be used for further research in non-EoE patients. Suggested reference: doi: 10.1080/00365521.2020.1831053

Response R1-10: As the reviewer’s suggested, an assessment using some immunohistochemical biomarkers based on the studies on EoE is promising to determine the diagnosis and appropriate treatments. Therefore, sentences” When viewed in a different light,…appropriate treatments [32].” (line 284-289 in revised version) in the limitation paragraph are now included.

Reviewer 2 Report

This communication is a combined case report and systematic review of the role of biologics in non-EoE EGID. It is informative but could be improved.

There are several areas where the English language does not seem to accurately capture the sentiment of the authors. Addressing these would significantly improve the manuscript.

For example: in line 29-30, dupilumab affects pathways not vice versa; lines 50-51, these biologics are not associated with type 2 inflammation, they target type 2 inflammatory pathways; line 54, I think it is too strong to think that these biologics are all EXPECTED to be effective against non-EoE EGIDs (hypothesize would be fine); line 85 is confusing.

Other areas:  Fig 2 and 3 could be improved - the food elimination arrow is not that helpful; there is lack of clarity of what the tissue Eos mean (which tissues?); 

The case report could be further clarified. My sense is that the patient's GI symptoms were controlled by diet, thus symptomatically the relevance of all the biologics are unclear.  Indicating that dupilumab improved food tolerance is not proven as this could have been part of the natural history of the disease and moreover re-introduction was apparently not attempted when on the two other biologics.  In Fig 3 it says OFC, I would suggest this is language best used for classic food allergy and not eosinophilic diseases which are chronic forms of Food allergy. 

Author Response

Dear Reviewer,

We thank reviewers for careful reading our manuscript and for giving us fruitful suggestions. Based on the reviewers’ comments, we have revised our manuscript entitled “Dupilumab Leads to Clinical Improvements Including the Acquisition of Tolerance to Causative Foods in Non- eosinophilic Esophagitis Eosinophilic Gastrointestinal Disorders” by Naoya Arakawa et al. I feel that the revised manuscript is significantly improved over the initial submission thanks to the reviewers’ suggestions. I would appreciate it if you would consider the manuscript for publication in Biomolecules.

Thank you in advance for your kind consideration of this paper.

Yoshiyuki Yamada

Point-by-point response for Reviewer 2 (R2)

This communication is a combined case report and systematic review of the role of biologics in non-EoE EGID. It is informative but could be improved.

There are several areas where the English language does not seem to accurately capture the sentiment of the authors. Addressing these would significantly improve the manuscript.

We thank the reviewer for giving me important and pertinent comments. We apologize that the English proofreading and word choice were insufficient.

Comment R2-1: For example: in line 29-30, dupilumab affects pathways not vice versa; lines 50-51, these biologics are not associated with type 2 inflammation, they target type 2 inflammatory pathways; line 54, I think it is too strong to think that these biologics are all EXPECTED to be effective against non-EoE EGIDs (hypothesize would be fine); line 85 is confusing.

Response R2-1: We changed the phase “be affected by” to “affect” in line 30 and the term” inflammation” to “pathway” in line 54 in revised version (no tracking version). Also, we modified the sentence as “Thus, we hypothesized that these biologics may be effective against non-EoE EGIDs.” at line 58 in revised version. About line 85 in original version, to be clear, we removed a phrase “from the published search criteria in the guidelines”.

Comment R2-2: Other areas: Fig 2 and 3 could be improved

Comment R2-2-1: The food elimination arrow is not that helpful.

Response R2-2-1: We removed the food elimination arrow in the figure.

Comment R2-2-2: There is lack of clarity of what the tissue Eos mean (which tissues?); 

Response R2-2-2: As there are no established diagnostic criteria for non-EoE EGID, gastrointestinal tissue eosinophil counts are helpful for diagnosis since physiological eosinophils are exclusively observed in gastrointestinal tract except for esophagus. Therefore, we used the criteria based on gastrointestinal tissue eosinophil counts in this study (reference 5). we added a sentence “Since GI eosinophil counts were high even if we assessed the tissue eosinophil count using recent criteria for non-EoE EGID in consideration of eosinophils physiologically present in the GI tract [26],”.

We showed eosinophil counts of stomach in timeline chart of the figure. Therefore, we modified Figure 2 legend in revised version (line 206).

Comment R2-3: The case report could be further clarified. My sense is that the patient's GI symptoms were controlled by diet, thus symptomatically the relevance of all the biologics are unclear.  Indicating that dupilumab improved food tolerance is not proven as this could have been part of the natural history of the disease and moreover re-introduction was apparently not attempted when on the two other biologics.  In Fig 3 it says OFC, I would suggest this is language best used for classic food allergy and not eosinophilic diseases which are chronic forms of Food allergy. 

Response R2-3: As the reviewer pointed out, re-introduction was not attempted during other biologics. However, she had occasionally presented symptoms due to accidental consumption of cow’s milk and hen’s egg during omalizumab and mepolizumab treatments. Therefore, the successful reintroduction was appeared to be due to the effect of dupilumab. We changed sentences as shown at line 194-197, “The successful reintroduction seemed to be due to the effect of dupilumab, since she had occasionally presented symptoms due to accidental consumption of cow’s milk and hen’s egg during omalizumab and mepolizumab treatments.”

We changed “OFC” to “re-introduction”.

Reviewer 3 Report

The authors have provided an interesting case report looking at the efficacy and clinical outcomes following treatments with 3 different biologics designated for allergic diseases in a single non-EoE EGID patient. The case report is combined with a mini systematic review of other reported literature on Biologic use in non-EoE EGIDs.

As the authors themselves state, it is relatively novel and rare, therefore not a lot of reports have been made perviously. Still, the study is valuable in kind to provide an overview of current state-of-knowledge and describes a single case report in depth. Moreover the literature reviewed is relevant and up-to-date. Overall a well-designed paper, with a nice flow of information that does not make exaggerated claims. 

(1) For ease of understanding the authors could combine Fig 1 and 2 into one figure, since it describes and follows a single patient over time. In that case the timeline could be (A), Eosinophil counts in tissue blood (B) and micrographs and endoscopic pictures as (C).

Table with histological findings (Fig 1B) could be separate.

(2) The paper could be further improved with a graphical abstract, that would attract reader´s interest.

Author Response

Dear Reviewer,

We thank reviewers for careful reading our manuscript and for giving us fruitful suggestions. Based on the reviewers’ comments, we have revised our manuscript entitled “Dupilumab Leads to Clinical Improvements Including the Acquisition of Tolerance to Causative Foods in Non- eosinophilic Esophagitis Eosinophilic Gastrointestinal Disorders” by Naoya Arakawa et al. I feel that the revised manuscript is significantly improved over the initial submission thanks to the reviewers’ suggestions. I would appreciate it if you would consider the manuscript for publication in Biomolecules.

Thank you in advance for your kind consideration of this paper.

Yoshiyuki Yamada

Point-by-point response for Reviewer 3 (R3)

The authors have provided an interesting case report looking at the efficacy and clinical outcomes following treatments with 3 different biologics designated for allergic diseases in a single non-EoE EGID patient. The case report is combined with a mini systematic review of other reported literature on Biologic use in non-EoE EGIDs.

As the authors themselves state, it is relatively novel and rare, therefore not a lot of reports have been made perviously. Still, the study is valuable in kind to provide an overview of current state-of-knowledge and describes a single case report in depth. Moreover the literature reviewed is relevant and up-to-date. Overall a well-designed paper, with a nice flow of information that does not make exaggerated claims. 

We thank the reviewer for giving us important and pertinent comments.

Comment R3-1: (1) For ease of understanding the authors could combine Fig 1 and 2 into one figure, since it describes and follows a single patient over time. In that case the timeline could be (A), Eosinophil counts in tissue blood (B) and micrographs and endoscopic pictures as (C).

Table with histological findings (Fig 1B) could be separate.

Response R3-1: Based on reviewer’s comments, we combined figure 2 and 3 and separated the table and micrographs. Since endoscopic pictures were chronological, we left them in the timeline chart. Concomitantly, clinical course of asthma exacerbations during biologics was separated as a Supplementary Figure (Supplementary Figure 1) since it was not directly associated with the main topic but helpful information.

Comment R3-2: (2) The paper could be further improved with a graphical abstract, that would attract reader´s interest.

Response R3-2: As the reviewer’s suggestion, we made a graphical abstract.

Round 2

Reviewer 2 Report

This revised manuscript is improved from the original and generally reads better, but there a few minor areas that could be improved to help the impact of the paper:

1. the new language around the paragraph on line 158 is confusing and should be modified

2. Line 183, consider changing since to "Of note,"

3. Line 199 is redundant with earlier language. It would be more helpful to specifically explain how omalizumab and mepolizumab, referred to in line 196, helped or did not help clinical symptoms, 

Author Response

We thank the reviewer for careful reading our manuscript and for giving us useful suggestions. Based on the reviewer’s comments, we have revised our manuscript entitled “Dupilumab Leads to Clinical Improvements Including the Acquisition of Tolerance to Causative Foods in Non- eosinophilic Esophagitis Eosinophilic Gastrointestinal Disorders” by Naoya Arakawa et al. We feel that the second revised manuscript is significantly improved over the first revision and the initial submission thanks to the reviewer’s suggestions. We would really appreciate if you consider the manuscript for publication in Biomolecules.

Thank you in advance for your kind consideration of this paper.

Yoshiyuki Yamada

Point-by-point response for Reviewer 2 (R2)

This revised manuscript is improved from the original and generally reads better, but there a few minor areas that could be improved to help the impact of the paper.

We thank the reviewer for giving us important and pertinent comments. Also, English proofreading of the changes in this revised version has been done by native English editing service.

Comment R2-1: The new language around the paragraph on line 158 is confusing and should be modified.

Response R2-1: We changed the sentence as follows; “However, the only cases with eosinophilic colitis (EoC) requiring colonoscopy for diagnosis were two patients in one study and our case.” (Line 158-161 in the revised version with tracking).

Comment R2-2: Line 183, consider changing since to "Of note,".

Response R2-2: We changed “Since” to “Of note,” (Line 184 in the revised version with tracking).

Comment R2-3: Line 199 is redundant with earlier language. It would be more helpful to specifically explain how omalizumab and mepolizumab, referred to in line 196, helped or did not help clinical symptoms. 

Response R2-3: We changed the phrases to explain how omalizumab and mepolizumab improved clinical symptoms as follows; “Anti-immunoglobulin E (IgE) antibodies and the suppression of the interleukin (IL)-5 signaling pathway improved the endoscopic findings and chronic digestive symptoms of non-EoE EGID; nevertheless, acute and subacute symptoms had occurred due to ingestion of the causative foods, meaning that the improvements were not clinically sufficient, and suppression of the IL-4/IL-13 signaling pathways led to the acquisition of tolerance to the causative foods.” (Line 197-203 in the revised version with tracking).